

# 1    Distribution and Sources of Air pollutants in the North

# 2    China Plain Based on On-Road Mobile Measurements

Yi Zhu[1], Jiping Zhang[2], Junxia Wang[1], Wenyuan Chen[1], Yiqun Han[1], Chunxiang Ye[3],
Yingruo Li[1], Jun Liu[1], Limin Zeng[1], Yusheng Wu[1], Xinfeng Wang[4], Wenxing Wang[4],
Jianmin Chen[4], and Tong Zhu[1,5*]
[1]State Key Joint Laboratory of Environmental Simulation and Pollution Control,
College of Environmental Sciences and Engineering, Peking University, Beijing
100871, China
[2]Institute of Atmospheric Physics, Chinese Academy of Sciences, Beijing 100029,
China
[3]School of Chemistry, University of Leeds, Leeds LS2 9JT, UK
[4] Environment Research Institute, School of Environmental Science and Engineering,
Shandong University, Ji'nan 250100, China
[5]The Beijing Innovation Center for Engineering Science and Advanced Technology,
Peking University, Beijing, China
*Corresponding Author: tzhu@pku.edu.cn





**Abstract.** The North China Plain (NCP) has been experiencing severe air pollution
problems with rapid economic growth and urbanisation. Many field and model studies
have examined the distribution of air pollutants in the NCP, but convincing results
have not been achieved mainly due to a lack of direct measurements of pollutants over
large areas. Here, we employed a mobile laboratory to observe the main air pollutants
in a large part of the NCP from June 11 to July 15, 2013. High median concentrations
of sulphur dioxide ($SO_2$) (12 ppb), nitrogen oxides ($NO_x$) ($NO+NO_2$; 452 ppb), carbon
monoxide (CO) (956 ppb), black carbon (BC; 5.5 μg m$^{-3}$) and ultrafine particles
(28350 cm$^{-3}$) were measured. Most of the high values, i.e., 95 percentile
concentrations, were distributed near large cities, suggesting the influence of local
emissions. In addition, we analysed the regional transport of $SO_2$ and CO, relatively
long-lived pollutants, based on our mobile observations together with wind field and
satellite data analyses. Our results suggested that, for border areas of the NCP, wind
from outside would have a diluting effect on pollutants, while south winds would
bring in pollutants accumulated during transport through other parts of the NCP. For
the central NCP, the concentrations of pollutants were likely to remain at high levels,
partly due to the influence of regional transport by prevalent south–north winds over
the NCP and partly by local emissions.
**Keywords:** North China Plain, Air pollution, Distribution, On-road mobile
measurements





## 1. Introduction

The North China Plain (NCP) is a geographically flat region in the northern part of
Eastern China, which includes Beijing, Tianjin, most of Hebei, Henan and Shandong
provinces, and the northern parts of Anhui and Jiangsu provinces. This region is
surrounded by the Yan Mountains to the north, the Taihang Mountains to the west and
the Bohai Sea to the east. The NCP covers an area of 300,000 km$^2$, which corresponds
to about 1/32 of the total area of China, but is home to approximately 1/5 of the
Chinese population. The NCP is the political, economic and agricultural centre of
China. Along with rapid economic growth and urbanisation, the NCP has been
experiencing severe air pollution problems (Donkelaar et al., 2010). On a global scale,
the NCP is a hotspot of nitrogen dioxide (NO$_2$), carbon monoxide (CO), sulphate and
particulate matter (PM) concentrations, according to both satellite observations and
model simulations (Chin et al., 1996; Yu et al., 2010; Bechle et al., 2011; Streets et al.,
2013; Bucsela et al., 2013). The concentrations of PM with an aerodynamic diameter
$\leq 2.5$ μm (PM$_{2.5}$) and PM$_{10}$ in the NCP are much higher compared to other rapidly
developing areas in China, such as the Yangtze River Delta (Hu et al., 2014).
According to the air-quality report published by the Chinese Ministry of
Environmental Protection, in 2013, 9 of the 10 most polluted cities in China were
located in the NCP. Severe pollution events occur frequently in this area. Therefore,
studies of air pollution problems in the NCP are essential to obtain general insights
into the unique patterns of air pollution in this area and for management of emissions
control policies by the government.



Over the past decade, there have been a number of investigations of air pollution in
the NCP taking advantage of observation sites, aircraft measurement platforms,
mobile laboratories, satellite data and air quality models. In the NCP, a network of
observation sites has been built for air pollution research, mostly located in and
around large cities, particularly Beijing (Xu et al., 2011; Xu et al., 2014; Wang et al.,
2013; Meng et al., 2009; Shen et al., 2011; Lin et al., 2011). Variability, sources,
meteorological and chemical impacts of air pollutants have been discussed by
analysing these observational results. The concentrations of long-lived pollutants have
been shown to be significantly influenced by wind, particularly the south and north
winds, indicating that regional transport plays an important role in urban air pollution.
In addition, model studies have yielded similar results in various areas of the NCP
(An et al., 2007; Zhang et al., 2008; Liu et al., 2013). Satellite data have indicated that
regional transport has a significant impact on the haze period in the NCP (Wang et al.,
2014). Thus, it is necessary to understand regional transport to address air pollution
problems in the NCP, which will require data on the distribution of air pollutants in
this region.
However, observational data from a single or limited number of measurement sites
cannot present the whole picture of air pollution in the NCP. A number of mobile
laboratory measurements (Johansson et al. 2008; Li et al., 2009; Wang et al., 2009;
Wang et al., 2011) and aircraft measurements (Huang et al. 2010; Zhang et al. 2011;
Zhang et al. 2009; Zhang et al. 2014) have been used to determine pollution
distributions mainly within the megacity of Beijing. There have been several reports





of model and satellite studies on the air pollution distribution in the NCP, or even the
whole of China (Wei et al., 2011; Zhao et al., 2013; Ying et al., 2014; Ding et al., 2015;
Ding et al., 2009). However, there are disagreements between these results, e.g.,
regarding the distributions of $NO_2$ in several hotspot areas produced from model
(CMAQ) and satellite (SCIAMACHY) measurements (Shi et al., 2008). Uncertainties
in model simulations such as emissions inventories, and in satellite measurements
such as the influence of clouds, can only be evaluated by measuring the spatial
distributions of pollutants over a large geographical area, which are still lacking.

In this study, we measured the concentrations of nitrogen oxides ($NO_x$), CO,

sulphur dioxide ($SO_2$), ultrafine particles and BC with a mobile laboratory platform in
the NCP. Satellite data and field wind measurements during the observation period
were also used. Our specific objectives were to collect a dataset showing the spatial
distribution of air pollutants in the NCP, and to characterise the regional transport
within and outside the NCP. This study was performed as part of the Campaigns of
Air   Pollution   Research   in   Megacity   Beijing   and   North   China   Plain
(CAREBeijing-NCP 2013), and involved comprehensive stereoscopic observations,
including observations of two super sites, several routine sites, mobile laboratories
and model work. This paper focuses on the distribution and transport of pollutants in
the NCP, mainly based on data collected from a mobile platform.



## 2. Experimental methods

### 2.1 Mobile laboratory and study area

A mobile laboratory was built by our research group, details of which were previously

described (Wang et al., 2009). Briefly, this mobile laboratory was constructed in 2006

on an IVECO Turin V diesel vehicle (L = 6.6 m, W = 2.4 m, H = 2.8 m; payload = 2.7

metric tonnes). Instrumentation was powered by two sets of uninterruptible power

systems (UPS), consisting of three series of 48 V/110 Ah lithium batteries, which

could support all of the equipment operations without interruption for up to 5 h. The

inlet systems for our mobile laboratories were specifically configured to

accommodate the type of measurement requirements and the instrument suite to be

employed in specific field campaigns.

Instruments deployed on the mobile laboratory included those for studying $NO_x$,

CO, $SO_2$, BC and ultrafine particles. $NO_x$ was measured using an $NO_x$ analyser with

an Mo-converter (Ecotech model 9841A; Ecotech, Knoxfield, Melbourne, Australia),

with a detection range of 0–500 ppb and uncertainty of 10 % at a time resolution of 30

s. CO was measured with a CO analyser by light absorption (Ecotech model 9830A)

with a detection range of 0–9.8 ppm at a time resolution of 40 s. $SO_2$ was measured

using an $SO_2$ analyser with a fluorescence cell (Ecotech model 9850A) with a

detection range of 0–221.3 ppb at a time resolution of 120 s. BC was measured using

a multi-angle absorption photometer (MAAP; Thermo model 5012; Thermo Scientific,

Waltham, MA), with a detection range of 0–20 $\mu g\ m^{-3}$. The online measurement data



from these instruments were recorded with an industrial personal computer.

Ultrafine particles were measured with a fast mobility particle sizer (FMPS, TSI

3090; TSI, Shoreview, MN), which covers particle sizes from 5.6 nm to 560 nm in 32
channels with a time resolution of 0.1 s. The data were recorded on a dedicated
computer. Other auxiliary data including temperature, relative humidity, barometric
pressure and GPS coordinates were also measured. The driving speed was kept stable
at around $100 \pm 5$ km h$^{-1}$ to cover as much distance as possible with the 5 h of power
supplied by the lithium batteries.

To establish the spatial distribution and characterise the regional transport of air

pollutants in the NCP, the routes for the mobile measurements were specially designed
to cover important emissions hotspots (Fig. 1) and to map large areas of the NCP. The
routes included the municipalities of Beijing and Tianjin, most of Hebei province, and
part of Shandong province, which is about 300 km wide from the west to the east and
400 km long from the north to the south, covering most of the NCP. To avoid traffic
jams and rough roads, only expressways were chosen for all routes. Limited by the
duration of battery power and the variability of boundary layer height, we could not
cover all routes in one trip. Instead, we divided the routes into five parts. Route 1 was
along the Taihang Mountains from Beijing to Shijiazhuang, located in the western part
of the NCP. Routes 2 and 3 were from Shijiazhuang to Dezhou and Cangzhou to
Baoding, respectively, which were generally located in central NCP. Routes 4 and 5
were from Tianjin to Beijing and around the south of Beijing, located in northern NCP.





In addition, we ran each route in one day. Two days were also needed for calibration
and maintenance of instruments. Therefore, it took one whole week to conduct a
single experiment. In total, six experiments, including one pre-test study, were
designed from June 1 to July 15, 2013. The pre-test study was conducted between
June 1 and June 7, and five formal repeated experiments were conducted between
June 11 and July 15 (Experiment 1 [E1], June 11–June 15; E2, June 17, June 18 and
June 20; E3, June 24–June 25; E4, July 2–July 7; E5, July 11–July 15). All trips were
started at about 09:00 and ended at about 14:00 to ensure that the boundary layer was
relatively stable during the observation period in 1 day. Unfortunately, data were
unavailable on several days because of heavy rain. The route design and trip runs are
shown in Figures 1 and 2, respectively.

Figure 1 here.

**2.2 The trajectories model**
A Lagrangian particle dispersion model, FLEXPART-WRF version 3.1 (Brioude et al.,
2013; Stohl et al., 1998; Stohl et al., 2005; Fast and Easter, 2006), was used to
determine the origin and transport pathways of the air mass arriving at the
vehicle-based mobile measurement laboratory. The wind field used to drive
FLEXPART was the time-averaged wind provided by the WRF with temporal
intervals of 10 min and horizontally spatial resolution of 2 km (The details of the
mesoscale meteorological model is described in S2.1). FLEXPART simulates the
transport and dispersion of tracers by calculating the backward trajectories of





multitudinous particles, which are termed plume back trajectories. In this model,
turbulence in the planetary boundary layer (PBL) is parameterised by solving the
Langevin equation, and convection is parameterised using the Zivkovic Rothman
scheme (Stohl et al., 2005). To improve the accuracy of the trajectory calculation, we
used high-resolution WRF simulation domain 4 outputs as the input meteorological
conditions for the FLEXPART model. The turbulence, convection and boundary layer
height were computed along the trajectories of tracer particles using the WRF output
data. Backward integration was performed every 5 min during the mobile observation
period in June 2013. For each integration, 2000 stochastic particles were released
initially from within a box $1 \times 1$ km$^2$ in horizontal extent and 1–50 m vertical height
above ground centred on the mobile measurement laboratory. A total of 2000 inert
tracer particles were released about every 5 min along the route of the vehicle. For
each release, the backward trajectories were simulated for at least 12 h, and the
particle locations were output every 10 min for analysis. The 12 h length of the
backward trajectories was chosen as a trade-off to adequately sample the history of
the air masses over the region of interest, while decreasing the trajectory error (Stohl,
1998; Zhang et al., 2012).
The footprints of backward trajectories were calculated to present plume
trajectories. Footprints in this context refer to the total residence times of released
particles, which were calculated following Ashbaugh et al. (1985) and de Foy et al.
(2009) by counting the accumulated number of particles during the integration within
each cell of a $2 \times 2$ km$^2$ grid. Various transport and diffusion patterns can well be





described by these footprints analyses (Zhang et al., 2012).
**2.3 Stationary measurement sites and the fire data**
Concentrations of air pollutants, including $NO_x$, $SO_2$, CO and BC, were measured
simultaneously at three stationary measurement sites during CAREBeijing-NCP 2013.
These were rural sites located at Gucheng, Hebei province (GC, 39.13°N, 115.67°E),
Quzhou, Hebei province (QZ, 36.78°N, 114.92°E) and Yucheng, Shandong province
(YC, 36.67°N, 116. 37°E) (Fig. 1). The GC stationary site was near route 1, and QZ
and YC stationary sites were near route 2.
The main pollutants at these sites were measured using commercial instruments. At
the QZ site, gas analysers were used to measure $NO_x$ (Ecotech model 9841A), CO
(Ecotech model 9830A) and $SO_2$ (Ecotech model 9850A). At GC and YC stations, gas
analysers were used to measure $NO_x$ (Thermo model 42C), CO (Thermo model 48i)
and $SO_2$ (Thermo model 48i), and BC was measured by MAAP (Thermo model

5012).

Fire data were obtained from the Moderate Resolution Imaging Spectroradiometer
(MODIS) installed in Terra and Aura. The territory passing times were 10:30 (local
time) and 13:30 (local time) for Terra and Aura, respectively. Fire images were
obtained from EOSDIS Worldview (NASA, https://earthdata.nasa.gov/labs/worldview

).





## 3. Result and discussion

### 3.1 Distribution of air pollutants

BC, $NO_x$, CO and $SO_2$ were measured on five routes during the experiment to
determine the concentrations of air pollutants on the routes and their spatial
distributions in the NCP. Figure 2 shows the results of our mobile measurements
obtained in 19 trips on the five routes from June 11 to July 15. The mean and median
concentrations of BC, $NO_x$, CO and $SO_2$ were 5.8 and 5.5 μg m$^{-3}$, 422 and 452 ppb,
1006 and 956 ppb and 15 and 12 ppb, respectively, in the whole study. These high
values were consistent with previous measurements of most pollutants at stationary
measurement sites in the NCP except for $NO_x$. For example, the measured
concentrations of $NO_x$, $SO_2$ and CO were $62.7 \pm 4.0$ ppb, $31.9 \pm 2.0$ ppb and $1990 \pm$
130 ppb in an urban site in the courtyard of China Meteorological Administration in
the Beijing area from November 17, 2007, to March 15, 2008 (Lin et al., 2011). These
values were 13–50 ppb, 5.7–30.3 ppb and 1100–1800 ppb at an urban site in Wuqing
(between Beijing and Tianjin) from July 9, 2009, to January 21, 2010 (Wu et al.,
2011); and 28.4 ppb, 17.2 ppb and 1520 ppb at the GC site from July 2006 to
September 2007 (Lin et al., 2009). In addition, the concentration of $NO_2$ measured at
the YC site from June 18 to June 30 was about 20 ppb (Wen et al., 2015). This
comparison with stationary site measurements suggested that our mobile
measurements reflected the heavily polluted conditions in the NCP, which ensured its
feasibility in profiling the distributions of these air pollutants.



Figure 2 here
The levels of CO, $NO_x$ and BC here were also comparable to those in previous
mobile laboratory measurements in European and American cities. Bukowiecki et al.
(2002) measured CO in Zürich, Switzerland, and the average concentration was about
600 ppb. Hagemann et al. (2014) measured $NO_x$ in Karlsruhe in Germany, and the
average concentration was about 20 ppb. In the USA, $NO_x$ was around 50 ppb in
Somerville (Padró-Martínez 2012), 200 ppb during rush hour in Boston (Kolb et al.,
2004) and ranged from 230 ppb to 470 ppb in Los Angeles (Westerdahl et al., 2005).
Padró-Martínez et al. (2012) also measured BC in Somerville, and reported average
concentrations of about 1 μg m$^{-3}$. As these measurements were obtained in
heavy-traffic areas in large cities, and our results were measured over a large region,
we concluded that the air pollution problems in the NCP are among the worst in the
world.
In contrast to these pollutants, a low concentration of $SO_2$ was consistently
measured throughout the whole study. The low levels of $SO_2$ could be attributed to the
Chinese government's effort to install desulphurisation devices in power plants and
major industrial sources since 1996.
As shown in Figure 3, the concentrations of BC, $NO_x$, CO and $SO_2$ were highly
variable on the different routes in the NCP. The concentration ranges of these four
species were 5–14 μg m$^{-3}$ for BC, 447–891 ppb for $NO_x$, 22.6–40.4 ppb for $SO_2$ and
1105–1652 ppb for CO. These extremely high values, i.e., 95 percentile



concentrations, were consistently found in various plumes near these emissions
hotspots in the NCP, which suggested a major influence on concentrations of
measured species in these hotspot areas by local emissions. The hotspots observed
here were mainly around the junction areas of our design routes, and they included but
were not limited to areas of Beijing, Tianjin, Baoding, Cangzhou, Dezhou,
Shijiazhuang and Zhuozhou. Previous model simulations and satellite measurements
in the NCP also confirmed the high concentrations of $NO_2$ around these large cities
(Shi et al., 2008). It is worth noting that these observed concentration hotspots moved
around the emissions hotspots, probably as a result of the varied transport processes in
different trips. For example, a pollution plume was detected 100 km to the south of
Cangzhou on June 20, but 130 km to the north of Cangzhou on July 6. In addition,
plumes were not always detected in different experiments around these cities, with the
exception of Shijiazhuang.

Figure 3 here.

During the five experiments, no clear temporal distributions of air pollutant
concentrations in the NCP were seen, except for the significantly low levels of $NO_x$
and $SO_2$ observed in the last experiments. However, no connections between the
decline in $NO_x$ and $SO_2$ concentrations and emissions or transport could be made. In
fact, the decline could probably be attributed to a wide range of precipitation that
occurred at that time.
In summary, our mobile laboratory measurements indicated spatial distributions of



the pollutants SO₂, CO and BC and the number density of fine particles. The
concentrations of air pollutants in the NCP were among the highest in the world and
extremely high concentrations were also observed around several cities.
**3.2 The influence of traffic emission**
The levels and distributions of air pollutants in the NCP are mainly attributable to
three sources, i.e., regional transport, local emissions and traffic emissions. On-road
measurements, however, could be greatly affected by traffic emissions (Wang et al.,
2009). The influence of traffic emissions on our mobile laboratory measurements is
discussed below.
According to the emissions inventories, vehicles were a considerable source of $NO_x$,
BC and CO. In the Beijing-Tianjin-Hebei area, the $NO_x$ (Annual Report of Chinese
Environmental Statistics [in Chinese], 2013), BC (Cao et al., 2006) and CO (Zhao et
al., 2012) emissions from vehicles were 30.5 %, 2.4 % and 20 % of total emissions,
respectively. SO₂ emissions from vehicles were negligible. Thus, the on-road
measurements of $NO_x$, BC and CO would have been influenced by traffic emissions
to various degrees. For example, mean and median values of $NO_x$ concentration were
$487 \pm 213$ ppb and 493 ppb in various routes in the first four experiments and $127 \pm$
100 ppb and 100 ppb even in the last experiment with the presence of wet deposition.
These on-road values were much higher than those observed in the monitoring sites in
surrounding cities. The mean $NO_x$ concentrations were about $11 \pm 6$ ppb measured on
June 11 at the GC site, $25 \pm 10$ ppb on June 12, $8.1 \pm 0.91$ ppb on June 18, $4.7 \pm 1.2$



ppb on June 25, $6.3 \pm 1.3$ ppb on July 3, $3.2 \pm 0.79$ ppb on July 12 at the QZ site, and
$13 \pm 4.1$ ppb on June 25 and $13 \pm 1.8$ ppb on July 3 at the YC site (Fig. 2).
In addition, a strong correlation ($r^2 = 0.99$) was found between on-road $NO_x$ and
NO (Fig. 4), with an average $NO/NO_2$ ratio of 4, which was much higher than the
value of 0.05–0.2 in the aged plumes (Finlayson-Pitts and Pitts, 2010). The results
indicated that $NO_x$ observed by our mobile laboratory was mostly influenced by fresh
vehicle emissions. Overall, the on-road $NO_x$ observations here were not representative
of the $NO_x$ levels in the NCP.

Figure 4 here.

For BC, CO and $SO_2$, the concentrations measured by the mobile laboratory and
nearby monitoring sites were comparable to some extent (Fig. 2). The BC
concentrations measured in nearby monitoring sites were $2.8 \pm 1.4$ μg m$^{-3}$ (June 12,
QZ), $4.9 \pm 0.72$ μg m$^{-3}$ (June 18, QZ) and $4.9 \pm 1.1$ μg m$^{-3}$ (June 25, YC). Compared
to stationary measurements, the BC concentrations measured by the mobile laboratory
were slightly higher, $4.8 \pm 2.2$ μg m$^{-3}$ on June 12, $6.8 \pm 2.3$ μg m$^{-3}$ on June 18 and 6.5
$\pm 3.3$ μg m$^{-3}$ on June 25. The CO concentrations measured at monitoring sites were
$1220 \pm 910$ ppb (June 12, QZ), $1000 \pm 140$ ppb (June 18, QZ), $730 \pm 210$ ppb (June 25,
YC) and $520 \pm 190$ ppb (July 3, YC). Similarly, the CO concentrations measured by
the mobile laboratory were $950 \pm 440$ ppb on June 12, $1030 \pm 530$ ppb on June 18,
$1020 \pm 680$ ppb on June 25 and $990 \pm 450$ ppb on July 3. The $SO_2$ concentrations
measured at monitoring sites were $10 \pm 2.4$ ppb (June 11, GC), $3.4 \pm 0.57$ ppb (July 11,



GC), $30 \pm 20$ ppb (June 12, QZ), $12 \pm 4.2$ ppb (June 25, QZ) and $10 \pm 4.6$ ppb (June
25, YC). Meanwhile, the $SO_2$ concentrations measured by the mobile laboratory were
higher in some trips and lower in others compared to the stationary measurements.
For example, lower $SO_2$ concentrations of $6.6 \pm 5.5$ ppb on June 11 and of $26 \pm 11$
ppb on June 12 were measured in these two trips, and higher $SO_2$ concentrations of
$7.1 \pm 2.9$ ppb on July 11 and of $27 \pm 16$ ppb on June 25 in other trips. In addition, the
concentrations of BC, CO and $SO_2$ were not correlated with those of NO. Traffic is
not a major source of atmospheric CO over the NCP region, as determined by
comparing CO column concentration from the satellite and traffic flux (Wu et al.,

2011).

Thus, unlike $NO_x$, gas pollutants including BC, CO and $SO_2$ were mainly affected

by sources, such as local emissions and transport, other than traffic emissions. The
mobile    laboratory    observations    reported    here    could    accurately    reflect    the
concentrations and spatial distributions of BC, CO and $SO_2$ in the NCP.
**3.3 The influence of regional transport**
Local emissions and regional transport are the two main sources of pollutants in the
NCP (Xu et al., 2011). As stated above, local emissions in large cities had a major
impact on the air quality in their adjacent areas. Regional transport also plays a major
role. Our study demonstrated that the contribution of regional transport could vary
both    spatially    and    temporally,    depending    on    a    number    of    parameters,    such    as
prevalent wind, terrain and vertical mixing. We also roughly divided the NCP into two





parts according to these parameters, i.e., the northern border area and the central area,
to discuss the influence of regional transport on air quality.
**3.3.1 The border areas of NCP**
The northern border area of the NCP included major parts of routes 4 and 5 and the
western border areas of the NCP included a major part of route 1. The area is
surrounded by the Taihang Mountains to the west and the Yan Mountains to the north.
The north wind prevailed in the winter and the south wind prevailed in the summer in
this area.
During the measurements, the three routes experienced both north and south winds.
Specifically, northwest winds and east winds brought outside air masses from
Northeast China and the Bohai Sea to the northern border area on July 2 and July 7,
respectively (Fig. 5). In both trips, the concentrations of $SO_2$, CO and BC were $4.5 \pm$
$2.3$ ppb, $550 \pm 240$ ppb and $5.0 \pm 2.6$ µg m$^{-3}$, respectively, on July 2 and $7.0 \pm 3.0$ ppb,
$1090 \pm 320$ ppb and $6.5 \pm 2.7$ µg m$^{-3}$, respectively, on July 7 (Fig. 2), which were the
lowest values observed here in the border areas of the NCP. These observations were
reasonable as areas including the Bohai Sea to the west, north and east of the NCP
were regions of low emissions and the clean air brought by northeast and east winds
could dilute the air pollutants in the border areas of the NCP.
Figure 5 here.
It is worth noting that the BC concentration was not lowest on July 2, which was



the opposite of the observations for the gas pollutants, $SO_2$ and CO. Satellite images
showed that there were many fire plots near route 1 on July 2 (Fig. S2). A featured
single peak of aerosol number density at around 50 nm (Fig. S3) further confirmed
that BC emissions from agricultural crop residue burning contributed significantly to
the BC levels on July 2 (Zhang et al., 2011; Hays et al., 2005; Li et al., 2007).
On June 24, June 14 and June 15, the air masses were transported inside the NCP
from the southern NCP to the northern border areas by south winds (Fig. 5). Under
these wind conditions, the concentrations of $SO_2$, CO and BC were $15 \pm 5.8$ ppb, 1300
$\pm 330$ ppb and $8.0 \pm 1.4$ μg m$^{-3}$, respectively, on June 24, $26 \pm 7.9$ ppb, $1200 \pm 230$
ppb and $6.5 \pm 1.5$ μg m$^{-3}$, respectively, on June 14 and $28 \pm 7.1$ ppb, $1600 \pm 370$ ppb
and $7.0 \pm 1.9$ μg m$^{-3}$, respectively, on June 15 (Fig. 2), which were among the highest
levels detected on these routes. According to the emissions inventories (Fig. 3), most
emissions hotspots were located in the central and southern parts of the NCP.
Pollutants could be easily accumulated in air masses of south and central NCP origin.
In addition, the Yan Mountains to the north of the border area stopped the possible
transport pathway of these air masses, which further enhanced the accumulation of
long-lived air pollutants in the northern border area of the NCP.
Although route 4 on July 6 was under the influence of south winds, as on June 14
(Fig. 5), the concentrations of $SO_2$ and BC on July 6 were $14 \pm 7.6$ ppb and $4.6 \pm 1.8$
μg m$^{-3}$, respectively, on July 6, which were much lower than the values of $26 \pm 7.9$
ppb and $6.5 \pm 1.5$ μg m$^{-3}$, respectively, on June 14. Meanwhile, the CO levels on these



two days were similar. One possible cause of the low concentrations of both $SO_2$ and
BC was the slightly higher boundary layer height on July 6 compared to June 14. The
precipitation that occurred on July 6, but not on June 14, appeared to be a more
important contributing factor. The solubility of CO is less than that of $SO_2$. Therefore,
the wet deposition lifetime of $SO_2$ would be much shorter, thus limiting the transport
distance of $SO_2$. Meanwhile, the wet deposition of BC particles would also prohibit its
long-range transport. This may explain the similar CO levels and low $SO_2$ and BC
levels on July 6 at the same time.

In conclusion, for the northern border area, local emissions and regional transport

from other NCP areas due to south winds were two main sources of long-lived
pollutants; both north and east winds had significant dilution effects on the
concentrations of gas pollutants. The wind dependency scatter plots for $SO_2$ were used
to show the contribution of regional transport to air pollution in the northern border
area of the NCP (Fig. 6). The results indicated that the high concentration was
connected to the south wind at a wind speed from 4 to 10 m $s^{-1}$. Similar results for
$SO_2$ and CO were reported for several sites in the northern NCP (Wu et al., 2011; Lin
et al., 2011; Lin et al., 2009). As the south wind usually prevailed in the summer and
north wind in the winter, the regional transport of long-lived pollutants within the
NCP from the central and southern parts to the northern parts should be prevalent in
the summer; while the dilution of air pollutants mainly by north winds and
occasionally east winds should be prevalent for the northern parts of the NCP in the
winter.



Figure 6 here.

### 3.2.2 The central NCP

The central NCP consisted of routes 2 and 3, where numerous heavily polluted cities
are located. The area was surrounded by the Taihang Mountains to the west or
emissions hotspots in other directions. While the north wind prevailed in the winter, as
for the northern border areas, low pressure prevailed in the summer with south and
northeast winds in this area.
During the observation period, the measurements along the two routes experienced
different wind fields, including southwest winds on June 12 and July 3, northeast
winds on June 18 and a low-pressure system with south and northeast winds on June
25 (Fig. S4). Unlike the northern border area of the NCP where strong north winds
had a dilution effect, the concentrations of gas pollutants were mostly high regardless
of the wind direction in the central NCP, e.g., on June 18 and July 3 (Fig. S4).
Generally, our observations were reasonable according to the unique terrain and
emissions map in central NCP. Due to the heavy emissions level in the central NCP
and surrounding areas, pollutants readily accumulated to high levels on their ways to
the central NCP in air masses from all directions, such as the clean air masses from
the Bohai Sea, West China and Northeast China, and polluted air masses from
Southeast China.
The situation was slightly different in areas along route 3, particularly for those off





the coast of the Bohai Sea. Route 3 experienced east winds on July 14 (Fig. S4), and
the concentrations of pollutants were low (Fig. 2). This was not only because of the
wet deposition from the rain on that day, but also the transport of clean air from the
Bohai Sea. A featured peak of aerosol number density at around 20 nm (Fig. S3)
further confirmed the incoming air from the Bohai Sea (Haaf and Jaenicke, 1980;
Hoppel et al., 1986).
Vertical mixing can also affect the concentrations of pollutants. For example, while
the wind fields were similar on June 13 and July 4 (Fig. 7), the concentration of
pollutants on June 13 was lower than that on July 4 (Fig. 2). This was because the
boundary layer was much higher on June 13 (976 m) than on July 4 (626 m), and the
strong vertical convection diluted the air pollutants.
Specifically, the relative contributions of emissions and regional transport to the
local air pollution levels were slightly different on different routes. At the junction of
routes 1 and 2 around Shijiazhuang area, the concentrations of air pollutants were
always high, except in the last experiment when wet precipitation occurred. The local
emissions contributed significantly to the air pollution levels in this area. The city of
Shijiazhuang is known as an emissions hotspot with heavy coal consumption.
Previous model result showed that Shijiazhuang is an important emissions hotspot of
$SO_2$ even in the whole NCP area (He et al., 2012). Meanwhile, the Taihang Mountains
to the west of the city prevented the diffusion of air pollutants. On the other hand,
transport convergence in front of the Taihang Mountains and Yan Mountains was



proposed in a previous study (Su et al., 2004), as a result of a low-pressure system
along the Taihang Mountains and Yan Mountains. Although the transport convergence
moved around in the north end, it always passed by the Shijiazhuang area. As shown
in Figure 3, high levels of pollutants, particularly relatively long-lived species such as
$SO_2$ and CO, were consistently observed at the western end of route 2 near
Shijiazhuang area. Broad peaks of $SO_2$ and CO concentrations, indicators of regional
transport plumes, were present near Shijiazhuang area (Fig. S5).
Similar to the situation in Shijiazhuang area, transport convergence would
occasionally pass through other cities along routes 2 and 3. In a typical case on June
13 (Fig. 7), air masses were transported far from the southwest of the NCP along the
transport convergence through Shijiazhuang and reached the area on route 3, with air
pollutants accumulating during transport and showing high concentrations.

Figure 7 here.

Overall, in most areas in central NCP, regional transport would play essential soles
in determining the local air pollution levels, although the underlying mechanisms
were different for the transport convergence area, central NCP area and coastal area.
The wind dependence scatter plots for $SO_2$ and CO were used as examples to show
the contribution of regional transport on air pollution in central NCP (Fig. 6). The
results indicated that the prevailing winds were southwest and northeast in central
NCP during our observation period. The concentrations of CO were independent of
wind direction and wind speed. In addition, the high concentrations of $SO_2$ were





related to southeast winds with high speed, which was about 5–10 m s$^{-1}$. This may
have been because of transport convergence. Due to the strong interaction of different
areas in central NCP, emissions control policies must consider the whole emissions
budget to achieve the air quality aims.

## 4. Conclusion

A mobile laboratory was employed to obtain snapshots of the spatial distributions of
air pollutants in the NCP. The concentrations observed were at the highest levels in
the world and were distributed unevenly in the NCP. Most high concentrations, i.e., 95
percentile concentrations, of air pollutants were found near emissions hotspots, which
suggested the influence of local emissions. However, regional transport of air
pollutants was also considered significant in determining the air quality in the NCP.
Back trajectory analysis, satellite data and tracer pollutants were combined to
recognise various cases of regional transport in both the northern border and central
NCP. Where the border areas would occasionally be diluted by winds from outside the
NCP, the central NCP was affected by regional transport of air pollutants with a few
exceptions, such as when precipitation occurred. To achieve the aims of air quality
locally, emissions control policies must consider the whole emissions budget in the
NCP.




The English in this document has been checked by at least two professional editors,
both native speakers of English. For a certificate, please see:
http://www.textcheck.com/certificate/bENNRx
**Data availability.** The data of mobile and stationary measurements are available upon
requests.
**Author contribution.** T. Zhu, Y. Zhu, Y. Han and W. Chen designed the experiments.
T. Zhu secured the research grants. Y. Zhu, Y. Han and W. Chen carried out the
experiments. J. Zhang developed the model code and performed the simulations. J.
Wang managed the data in the program. J. Liu provided the emission maps. L. Zeng,
Y. Wu, X. Wang, W. Wang and J. Chen provided the data of stationary measurements.
Y. Zhu analyzed the data with contributions from all co-authors. Y. Zhu prepared the
manuscript with helps from T. Zhu, C. Ye and Y. Li.
**Acknowledgement.** This study was supported by the National Natural Science
Foundation Committee of China (21190051, 41121004, 41421064), the European 7th
Framework Programme Project PURGE (265325), the Collaborative Innovation
Center for Regional Environmental Quality.

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





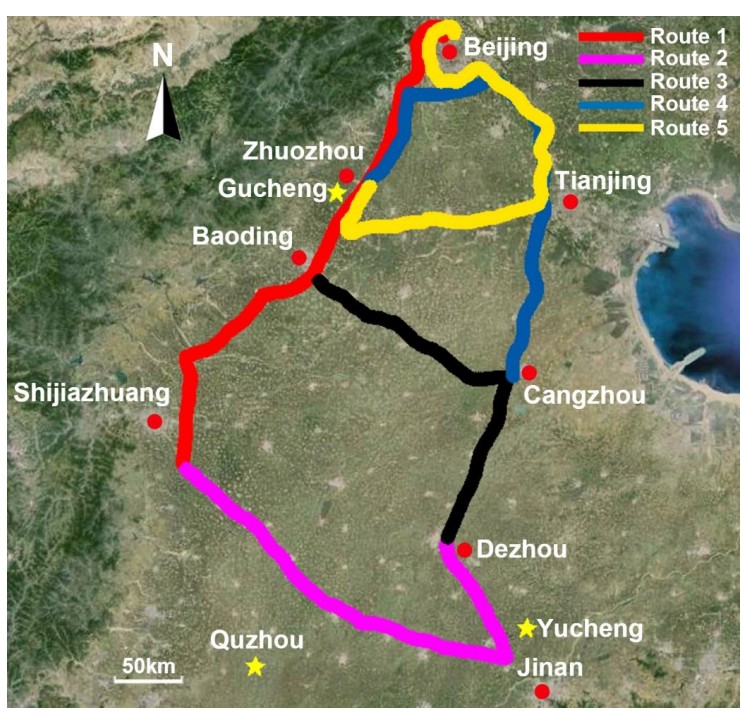


**Fig. 1.** The study area in NCP. The red track shows route 1, from Beijing to Shijiazhuang. The

purple track shows route 2, from Shijiazhuang to Dezhou. The black track shows route 3, from

Dezhou to Baoding. The blue track shows route 4, from Cangzhou to Zhuozhou. The yellow track

shows route 5, from Zhuozhou to Beijing. The red round dots on the map present the major cities

near the routes. The yellow five-pointed stars present the monitoring sites.




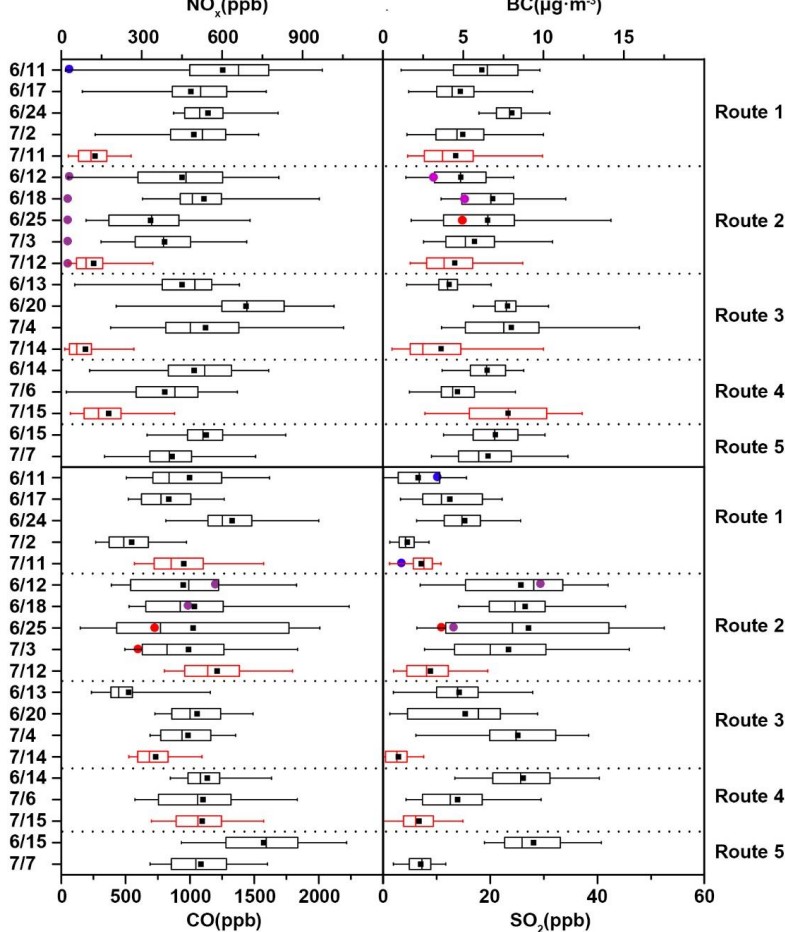

696

**Fig. 2.** The concentrations of SO₂, CO, BC and NO$_x$ in each trip in different routes. The red boxes

were the results in the last experiment. Values marked were the 5[th] and 95[th] percentile (-), standard

deviation (lower and upper box lines), median (middle box line), and mean (■). The blue dots

were results of GC station. The purple dots were results of QZ station. And the red dots were

results of YC station.



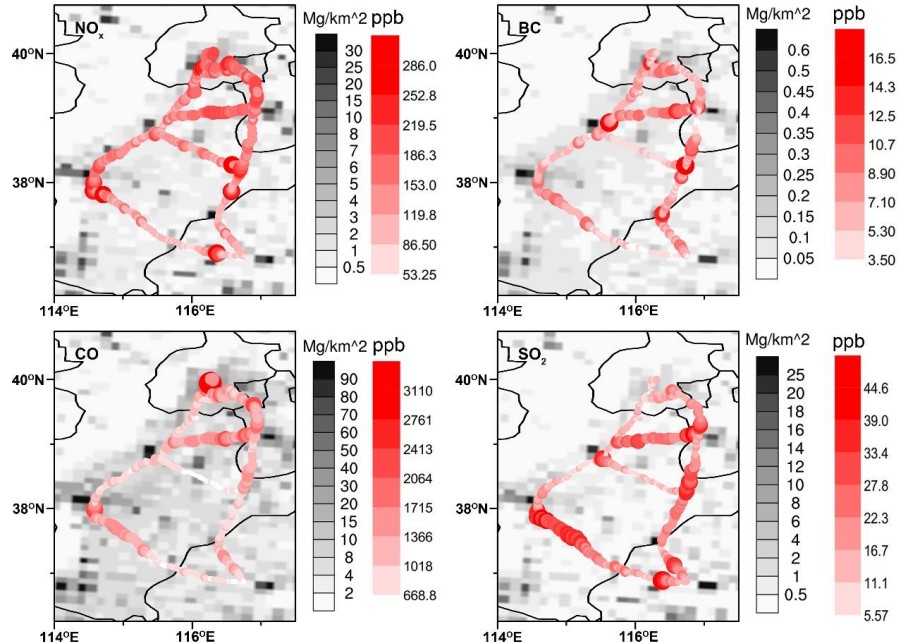

702

**Fig. 3.** The spatial distributions of the measured concentrations in our study and the emission

maps of SO₂, NOₓ, CO and BC. The colored tracks were average concentrations measured in this

mobile observation. The black and white maps were emission maps of the year 2010 derived from

MEIC model (Zhao et al, 2013).





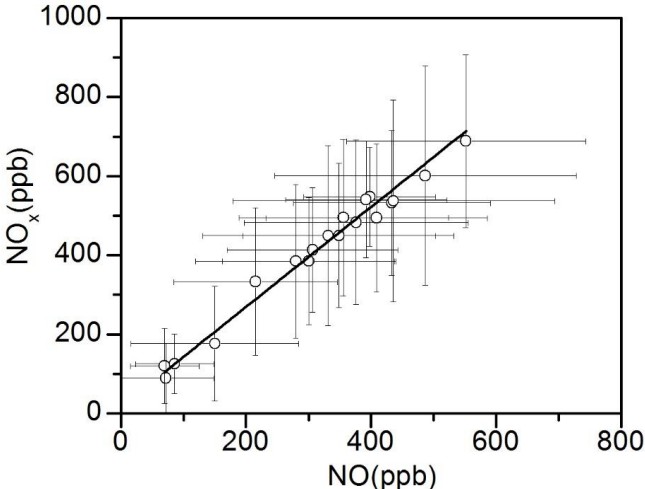


**Fig. 4.** The regression curve of the means of concentrations of $NO_x$ and NO and the error bars in

all 19 trips.

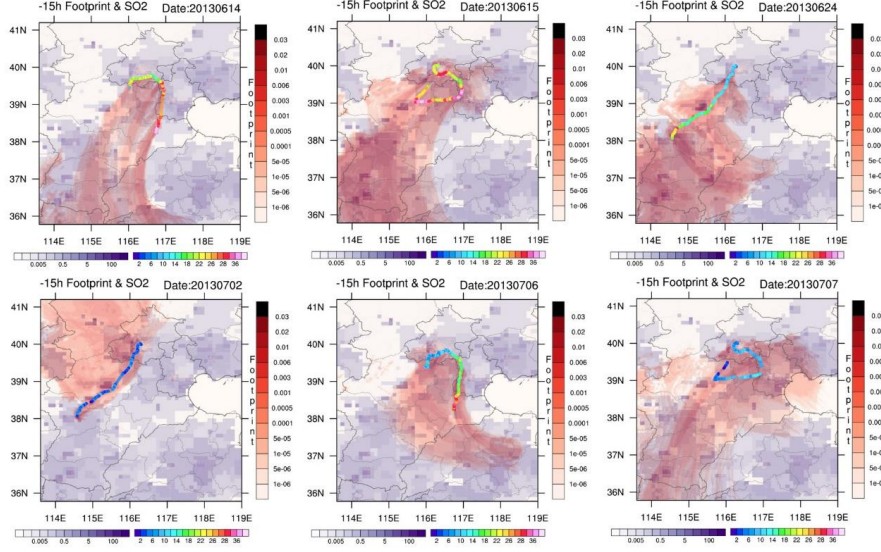


**Fig. 5.** The back trajectories of observed air masses in the borders of NCP in June 14, June 15,
June 24, July 2, July 6 and July 7.





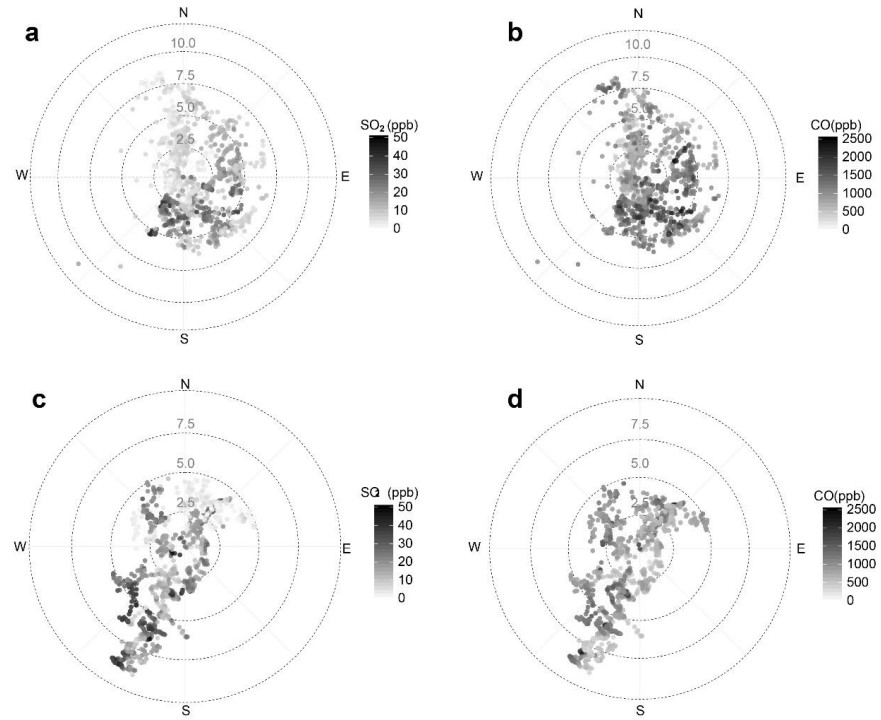


**Fig. 6.** Wind dependency scatter plots of concentrations of SO₂ and CO in border and central areas

in NCP (a. SO₂ in border area; b. CO in border area; c. SO₂ in central area; d. CO in central area).

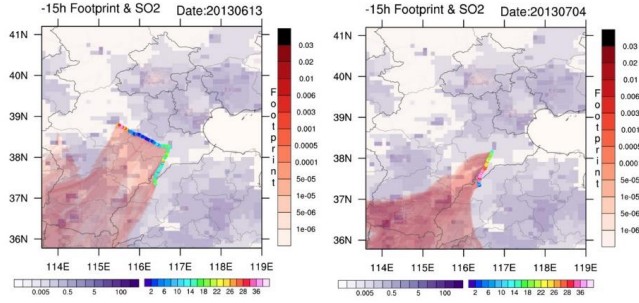


**Fig. 7.** The back trajectories of observed air masses in the central NCP in June 13 and July 4.