# Peer review of "Distribution and Sources of Air pollutants in the North"

_Atmospheric Chemistry and Physics, 2016_

## Referee Comment (RC1) · Anonymous Referee #1 · 15 Jun 2016

This manuscript reports measurements of SO2, NOx, CO and black carbon made in a mobile van traveling on five expressways in the North China Plain in summer 2013. The authors offered some general discussions on sources and long-range transport of these pollutants. On-road measurements are normally used to understand emission characterises of road traffic, but this appears not the case for this study which attempts to study the spatial distributions of the air pollutants in the NCP region. I doubt this objective can be achieved due to potentially large impact from vehicle emissions on the data. Another concern is that the reported data may have major flaws. The extremely high NOx values are not consistent with moderate values of the other three pollutants. If the NOx data are correct, they (mean value=452 ppbv) clearly show huge impact of on-road vehicles on the measurements. However, the mean CO value is only ∼ 1 ppm, which seems too low. Is this due to the dominance of diesel vehicles on the

highways? If so, the measured black carbon would be significantly affected too by the diesel vehicles. How were the instruments calibrated? Did you make measurements off road to compare with the on-road data to check the impact of road vehicle emissions? In summary, the authors are advised to clarify these two important issues (the intended use of the on-road measurements and the data quality). In addition, the analysis and discussion of the data set should be more in-depth.

---

## Author Comment (AC1) · 10 Jul 2016

We thank the reviewer for the critical comments, which are very helpful in improving the quality of the manuscript. We are revising the manuscript according to the suggestion and the following lists our responses to the comments.

Referee Comment #1:

This manuscript reports measurements of SO2, NOx, CO and black carbon made in a mobile van traveling on five expressways in the North China Plain in summer 2013. The authors offered some general discussions on sources and long-range transport of these pollutants. On-road measurements are normally used to understand emission characterises of road traffic, but this appears not the case for this study which attempts to study the spatial distributions of the air pollutants in the NCP region. I doubt this

objective can be achieved due to potentially large impact from vehicle emissions on the data. Another concern is that the reported data may have major flaws. The extremely high NOx values are not consistent with moderate values of the other three pollutants. If the NOx data are correct, they (mean value=452 ppbv) clearly show huge impact of on-road vehicles on the measurements. However, the mean CO value is only ∼1 ppm, which seems too low. Is this due to the dominance of diesel vehicles on the highways? If so, the measured black carbon would be significantly affected too by the diesel vehicles. How were the instruments calibrated? Did you make measurements off road to compare with the on-road data to check the impact of road vehicle emissions? In summary, the authors are advised to clarify these two important issues (the intended use of the on-road measurements and the data quality). In addition, the analysis and discussion of the data set should be more in-depth.

Response: 1. In previous studies, on-road mobile measurements were not only used for estimating the vehicular emissions, but also for analyzing the spatial distributions and the relationship between cities and regions (Kolb et al., 2004; Johansson et al., 2008; Wang et al., 2011).

The influences of vehicular emissions on on-road mobile measurements were different for NOx, CO and SO2. According to the analyzation in section 3.2 in the paper, NOx was mainly from vehicular emissions. According to the reports of SO2 emission inventory (Li et al., 2015) and vehicular emission inventory (Cai and Xie, 2007), the contribution of vehicular emission to SO2 was not significant during the time of our measurements.

Based on the monitoring results along the highways in our study area (Zhang et al., 2003), the ratio of the amount of gasoline and diesel vehicles was about 1:1. During our measurements, we found that the diesel vehicles were mainly diesel buses and median-duty diesel trucks. With the reported on-road diesel vehicular emission factors for CO and NOx in China (Table 1) (Shen et al., 2015), we calculated that the overall vehicular emission factors for CO and NOx during our measurements were 3 g/km and

5 g/km. The concentration of NOx on road was about 400 ppb higher than that off road. Thus, it is estimated that the concentration of CO on road was about 240 (=400/5*3) ppb higher than that off road, which contributed about 24% of the total concentration. It can also be found from the temporal distributions of concentrations of CO, NOx and BC, e.g. in June 13 (Fig. 1), that the concentration of CO on road (482 ppb) was about 25% higher than that of off-road (391 ppb).

In conclusion, vehicular emissions contributed less than 30% of CO measured in this experiment, and the CO and SO2 measurement results could be used for studying the spatial distributions of the air pollutants in the NCP region.

2. In previously reported studies of on-road mobile measurements, the daily concentration of NOx could be higher than 400 ppb. Considering that the concentration of CO on road was about 25% higher than that of off-road, and the background concentration of CO in the North China Plain was 0.4-0.7 ppm, the result of CO measured in this experiment was reasonable. Besides, the concentrations of NOx and BC measured had similar trends (Fig. 1) and correlated each other well (Fig. 2). Based on this regression curve, the ratio of emission factors of BC and NOx was 0.004:1, which was close to the previously reported diesel vehicular emission factors in Beijing (0.008:1) (Wang et al., 2012). It could be conclude that the diesel vehicular emissions had a large impact on the BC and NOx concentrations measured in NCP.

We have carried out quality assurance and quality control of the on-road measurements, including instruments calibrations and inter-comparison with monitoring station in the campus of Peking University (Wang et al., 2009). One of the instruments calibrations results is shown in Figure 3.

Reference

Cai, H. and Xie, S.: Estimation of vehicular emission inventories in China from 1980 to 2005, Atmos. Environ., 41, 8963-8979, 2007.

Johansson, M., Galle, B., Yu, T., Tang, L., Chen, D., Li, H., Li, J., and Zhang, Y.: Quantification of total emission of air pollutants from beijing using mobile mini-doas, Atmos. Environ., 42, 6926–6933, 2008.

Li, Q., Cao, G., Liang, D., and Lyu, Y.: The emission inventories prediction of SO2 and NOx in China mainland based on scenario analysis, Environmental Pollution & Control, 37, 9-19, 2015.

Kolb, C., Herndon, S., Mcmanus, J. B., Shorter, J., Zahniser, M., Nelson, D., and Jayne, J., Canagaratna, M. R., and Worsnop, D. R.: Mobile laboratory with rapid response instruments for real-time measurements of urban and regional trace gas and particulate distributions and emission source characteristics, Environ. Sci. Technol. 38, 5694-5703, 2004.

Shen, X., Yao, Z., Zhang, Q., Wagner, D. V., Huo, H., Zhang, Y., Zheng, B., and He, K.: Development of database of real-world diesel vehicle emission factors for China, J. Environ. Sci., 31, 209–220, 2015.

Wang, M., Zhu, T., Zhang, J., Zhang, Q., Lin, W., Li, Y., and Wang, Z.: Using a mobile laboratory to characterize the distribution and transport of sulfur dioxide in and around Beijing, Atmos. Chem. Phys., 11, 11631–11645, 2011.

Wang, M., Zhu, T., Zheng, J., Zhang, R., Zhang, S., Xie, X., Han, Y., and Li, Y.: Use of a mobile laboratory to evaluate changes in on-road air pollutants during the Beijing 2008 Summer Olympics, Atmos. Chem. Phys., 9, 8247–8263, 2009.

Wang, X., Westerdahl, D., Hu, J., Wu, Y., Yin, H., Pan, X., and Zhang, K.: On-road diesel vehicle emission factors for nitrogen oxides and black carbon in two Chinese cities, Atmos. Environ., 46, 45-55, 2012.

Zhang, W., Wang, W., Niu, X., and Hu, G.: Comparison analysis of vehicle fuel consumption on Jing-Shi Expressway and its corresponding parallel road, Journal of highway and transportation research and development (In Chinese), 20, 182-185, 2003.

[Figure]

[Figure]

**Fig. 1.** Temporal distributions of concentrations of CO, NOx and BC in June 13.

$$y = 0.0051x + 1.8385$$
$$R^2 = 0.4213$$

BC(ug m$^{-3}$)

NO$_x$(ppb)

**Fig. 2.** The regression curve of the concentrations of NOx and BC in June 13.

[Figure]

[Figure]

**Fig. 3.** Calibration curves of gas analyzers in June 16.

| Vehicle type | Diesel bus | Median-duty diesel truck | Light-duty gasoline vehicles |
|---|---|---|---|
| CO | 1.2 g/km | 1.5g/km | 4.4 g/km |
| NOx | 11 g/km | 6.4 g/km | 1.3 g/km |

**Fig. 4.** On-road diesel vehicular emission factors for CO and NOx in China (Shen et al., 2015).

---

## Referee Comment (RC2) · Anonymous Referee #2 · 22 Jul 2016

Y. Zhu et al., Distribution and sources of air pollutants in the North China Plain based on on-road mobile measurements, for ACP

The authors have conducted five campaigns (plus an earlier pre-test) along identical routes across the North China Plain (NCP) and measured NOx, CO, SO2, ultrafine particles and black carbon on a mobile laboratory. The route covered a nice perimeter of the NCP and at least two roughly east-west transects. The mobile sampling was conducted on expressways from 0900-1400 each day. The authors discuss the observations in the context of wind/trajectories (e.g. from cleaner regions such as the mountains or sea). Overall, the manuscript is well-written.

My concern with this manuscript is that the analyses of the measurements, by the very nature of mobile laboratory measurements without supporting other measurements

such as aircraft, ground-based networks, satellite, etc., are quite limited. Specific comments:

1. Most seriously, I have concerns about the representativeness of the dataset, given that the measurements were only on expressways. Just because vehicle emissions aren't major sources in the regional inventory, they are very important when sampling in the aggregate exhausts of other vehicles. Therefore, arguments stating that they are small regionally do not hold for these measurements. Indeed, the NOx values are argued as being influenced by vehicles – why wouldn't SO2, CO, and black carbon be impacted? CO has a two month lifetime on average, yet measurements of CO on a highway are typically much higher than, say, at a park 1 km away. It is the enhancements above the regional background that matter, and trying to separate local (road) vs. regional background measurements is not clear without extensive, other data. Did the authors drive off-expressway to examine how their measurements changed when off-expressway and (ideally) away from other vehicles (parks)? How much traffic was around the mobile laboratory? Were higher concentrations reported while in traffic jams (surrounded by cars)? The extent to which the mobile vehicle measures "expressway" versus more "regional background" levels will vary depending on the traffic density, winds, and stability. Consider the two extreme cases: 1) on a road with no other vehicles, the mobile lab will measure the regional background; 2) while stuck in traffic in a multi-lane city expressway, the measurements are clearly just vehicle emissions. How this measurement footprint changes versus location/day/traffic/wind is difficult to quantify without more measurements (see comment 3 below for more details). WFR-CHEM or FLEXPART can't model at roadway scales, of course, so there has to be some criteria applied to assess such influences. On p. 7, it is stated that traffic jams were avoided – this is rather ambiguous. How was this defined? In all this driving, there was never a traffic jam?

2. A large number of values are reported, and too often this reads as a "data dump" with little insight besides vague generalizations. Tables comparing select case studies

may be helpful, particularly highlighting the different spatiotemporal domains of the comparisons.

3. Some comparisons to ground-based sites were noted (also to literature), but even these were rather trivial and limited. Comparing a transect along a nearby road to a nearby stationary site (e.g. "YC" in text), it is unclear if the mobile laboratory stopped there for a certain length of time or if it just passed by the site as part of the drive. How close to the site was the vehicle located and for what time domain for the comparisons listed (e.g. Fig. 2, p15, lines 300-310, etc.)? By knowing the mean wind, one should be able to link space-time scales at some level. Instead, just a date is given – did the authors integrate for the entire day for both measurements? Given the contamination / variable contribution of vehicle exhaust in their data set, quantifying agreement is frought with uncertainties.

4. As the manuscript noted, concentrations are impacted by wind direction (source regions upstream), local sources (which are never really specified – just stating "urban" area), location (e.g. in the central region wind direction plays less of an influence than at the margins) and boundary layer height (e.g. p. 21, line 421-425). It is hard to interpret the variations observed in Fig. 2 – which are caused by which of the above factors and which ones dominate (and when)? Reasonable examples are provided in the following text of all of the above, but in the end, I have difficulty interpreting the importance of each of these processes in the full dataset.

Other: - Experimental methods: what are the precisions and uncertainties in the measurements (e.g. NOx precision; CO precision and accuracy; SO2 precision and accuracy; BC same thing), only NOx had an uncertainty noted (but no precision). Did the laboratory studies/calibrations done on non-driving days agree or disagree with stated manufacture specifications?

- To what extent was the data coverage for the 5 campaigns? 100%? Rain is mentioned on p. 8, line 152 – what about other data dropouts (if they existed)?

- p. 12, "low levels of SO2": 10 ppbv SO2 is not low, perhaps compared to the past in China. But I'm not sure how this mobile lab study over a very limited time/space can verify that SO2 has decreased and confirmed policy desulphurization devices. No comparison to past SO2 measurements in these locations is noted, so this seems unsubstantiated from this study.

- Some data should be made available to the community on a website, not "on request", given the political sensitivity of Chinese air pollution data. What metadata will be provided, e.g. GPS, vehicle information (speed), met data, etc.?

- I like the FLEXPART trajectory attribution, but again this is a qualitative comparison as the upwind values were not measured. So while it makes sense that northerly winds off the mountains will decrease values nearby, I don't think this was particularly profound.

Overall: The limitations of mobile laboratory deployments by themselves are apparent in this study. Unless the authors can address some the ambiguities noted above in a quantitative way (or refocus onto vehicle emissions, or restrict data analyses/measurements to certain conditions that are representative of regional backgrounds), I'm struggling to see how this study adds significantly to the literature in a journal such as ACP.

---

## Author Response (AR1)

**Responses to the Comments of the Referees**

**Distribution and Sources of Air pollutants in the North China Plain Based on On-Road Mobile Measurements**

Yi Zhu, Jiping Zhang, Junxia Wang, Wenyuan Chen, Yiqun Han, Chunxiang Ye, Yingruo Li, Jun Liu, Limin Zeng, Yusheng Wu, Xinfeng Wang, Wenxing Wang, Jianmin Chen, and Tong Zhu

We thank the reviewers for the critical comments, which are very helpful in improving the quality of the manuscript. We have made major revision based on the critical comments and suggestions of the reviewers.   Our point-by point responses to the comments are listed in the following.

**Referee #1:**

***Comment No. 1***. *This manuscript reports measurements of $SO_2$, $NO_x$, CO and black carbon made in a mobile van traveling on five expressways in the North China Plain in summer 2013. The authors offered some general discussions on sources and long-range transport of these pollutants. On-road measurements are normally used to understand emission characterises of road traffic, but this appears not the case for this study which attempts to study the spatial distributions of the air pollutants in the NCP region. I doubt this objective can be achieved due to potentially large impact from vehicle emissions on the data.*

**Response:** Actually in previous studies, on-road mobile measurements have also been used for analyzing the spatial distributions (Kolb et al., 2004), the urban emissions (Johansson et al., 2008), and the regional transport of air pollutions (Wang et al., 2011).

We agree that we need to provide more evidences to justify the representativeness of our dataset for the regional background concentrations of air pollutants. Following the comments and suggestions of the reviewers, we have rewritten the manuscript and provided the following evidences:

1. **Comparing the on- and off-road measurements of air pollutants to estimate the enhancement of air pollution on highway above the regional background**

Each on-road measurement trip started from a parking lot in a highway service center and ended at the parking lot of another service center. The parking lots are about 150 m away from highway, such as those in service centers Dezhou (DZ) and Xizhaotong (XZT) (Fig. 1). Using the difference of the concentrations of air pollutants measured in a parking lot and on highway, we can estimate the level of the enhancement of the concentrations of air pollutants on highway above regional backgrounds.

[Figure]

Fig. 1. The locations of the mobile platform parked at the parking lots in the service centers Xizhaotong (XZT) and Dezhou (DZ), and their distances to high ways (from Google Map).

Table 1 shows the 5 min averaged concentrations of $NO_x$, CO, $SO_2$, and BC measured in parking lots and on highways, and the difference between the on- and off-road concentrations. The concentrations in parking lots were measured for 5 mins before driving or after parking with engine turned off, and the concentrations on highways were measured for 5 mins after entering highways or 5 mins before entering the service centers.

The concentrations of $NO_x$ measured on highway show drastic enhancement than those measured off-road, from 19 to 449 ppb, or 43% to 1658% (510±61%, mean± Standard deviation), while other pollutants have much lower enhancement or even reduction in concentrations. The difference between the on-road and off-road concentrations of CO ranged from -478 to 145 ppb, or -28% to 34% (7±22%); for $SO_2$, it is -26 to 13 ppb or -18% to 175% (52±59%); for BC, it is -0.7 to 4.7 µg m$^{-3}$ or -11% to 261% (85±90%).

The 175% enhancement of $SO_2$ on July 12 could be due to the much lower $SO_2$ concentration, 1.2 to 3.3 ppb, than those measured in other days. The rain on July 12 is likely the major reason for the much lower concentrations of $SO_2$ (3 ppb) and $NO_x$ (63 ppb), while CO and BC, which are much less water soluble, show no significant difference than those measured in other days. If we ignore the results of $SO_2$ and $NO_x$ on July 12, then the difference between the on- and off-road concentrations were 82% to 1658% (510±61%) for $NO_x$ and were -18% to 87% (31±31%) for $SO_2$.

Apparently, vehicular emission is the major source lead to the 82% to 1658% enhancement of $NO_x$ concentrations on highway. The enhancement of CO and $SO_2$ concentrations on highway were mostly less than 30%, suggesting vehicular emission is not the main source for CO and $SO_2$ on highway, regional background is the dominant factor determine their concentrations.

Table 1. The concentrations of $NO_x$, CO, $SO_2$ and BC measured in parking lots and high ways, and the concentration differences between off-road and on-road measurement.

| Date | Parking lot | Concentrations | | | | | | | |
|------|-------------|----------------|--|--|--|--|--|--|--|
| | | $NO_x$ (ppb) | | | | CO (ppb) | | | |
| | | Off-road (C1) | On-road (C2) | C2-C1 | $\dfrac{C2-C1}{C1}$ | Off-road (C1) | On-road (C2) | C2-C1 | $\dfrac{C2-C1}{C1}$ |
| 6/11 | XZT | 234 | 643 | 409 | 175% | 694 | 677 | -17 | -2% |
| 6/12 | XZT | 411 | 699 | 288 | 70% | 1050 | 1098 | 48 | 5% |
| 6/12 | DZ | 20 | 106 | 86 | 430% | 396 | 506 | 110 | 28% |
| 6/13 | DZ | 19 | 334 | 315 | 1658% | 431 | 576 | 145 | 34% |
| 6/17 | XZT | 30 | 297 | 267 | 890% | 1000 | 1086 | 86 | 9% |
| 6/18 | XZT | 109 | 468 | 359 | 329% | 1212 | 955 | -257 | -21% |
| 6/18 | DZ | 372 | 677 | 305 | 82% | 588 | 711 | 123 | 21% |
| 7/3 | XZT | 49 | 498 | 449 | 916% | 826 | 861 | 35 | 4% |
| 7/12 | XZT | 44 | 63 | 19 | 43% | 1708 | 1230 | -478 | -28% |

Table 1 (continuous). The concentrations of NOx, CO, $SO_2$ and BC measured in parking lots and high ways, and the concentration differences between off-road and on-road measurement.

| Date | Parking lot | Concentrations | |
|------|-------------|----------------|--|
| | | $SO_2$ (ppb) | BC ($\mu$g m$^{-3}$) |

|  |  | Off-road (C1) | On-road (C2) | C2-C1 | $\frac{C2-C1}{C1}$ | Off-road (C1) | On-road (C2) | C2-C1 | $\frac{C2-C1}{C1}$ |
|---|---|---|---|---|---|---|---|---|---|
| 6/11 | XZT | 11.3 | 12.7 | 1.4 | 12% | 2.3 | 5.6 | 3.3 | 143% |
| 6/12 | XZT | 24.4 | 30.9 | 6.5 | 27% | 2.9 | 6.6 | 3.7 | 128% |
| 6/12 | DZ | 8.9 | 10.0 | 1.1 | 12% | 1.5 | 4.1 | 2.6 | 173% |
| 6/13 | DZ | 14.6 | 12 | -2.6 | -18% | 2.3 | 3.6 | 1.3 | 57% |
| 6/17 | XZT | 11 | 17 | 6 | 55% | 3.4 | 5.5 | 2.1 | 62% |
| 6/18 | XZT | 15 | 28 | 13 | 87% | 6.6 | 5.9 | -0.7 | -11% |
| 6/18 | DZ | 17 | 21 | 4 | 24% | 7.4 | 10.3 | 2.9 | 39% |
| 7/3 | XZT | 17 | 21.9 | 4.9 | 29% | 1.8 | 6.5 | 4.7 | 261% |
| 7/12 | XZT | 1.2 | 3.3 | 2.1 | 175% | 5.3 | 6.2 | 0.9 | 17% |

**2. Time series of the concentrations of air pollutants measured on highway**

Figure 2 shows the concentrations of $NO_x$, CO, $SO_2$, and BC measured on highway on a typical day, June 13, 2013. Apparently, the concentration of BC follows the trend of $NO_x$ concentration and those of CO and $SO_2$ did not. This suggests the high enhancement of $NO_x$ and BC concentration on highway was due to the vehicular emission, while the concentrations of CO, $SO_2$ were not.

[Figure]

Fig. 2. The time series of the concentrations of $NO_x$, CO, $SO_2$ and BC measured on June 13, 2013.

**3. Using the ratios of weighted vehicular emission factors to estimate the concentration enhancement of $NO_x$, CO, $SO_2$ and BC measured on highway**

Based on the reported vehicular emission factors (Shen et al., 2015; Cai and Xie, 2007, 2010; Lei et al., 2011) and the vehicle composition (Chinese Automotive Technology & Research Centre, 2015) in Hebei province, where we conducted the most of the mobile measurements, we estimated the weighted vehicular emission factors on the highways (Table 2). The factors for $NO_x$, CO, $SO_2$, and BC during our measurements were 2.9 g km$^{-1}$, 4.8 g km$^{-1}$, 0.04 g km$^{-1}$, and 0.01 g km$^{-1}$, respectively. If we assumed 400 ppb as the $NO_x$ concentration enhancement on-road caused by vehicular emission, using the ratios of the weighted vehicular emission factors, the estimated enhancements of CO, $SO_2$ and BC concentrations on-road emitted by vehicles were 240 ppb, 3 ppb and 1 μg m$^{-3}$. These are at the similar levels of those enhancement showed in Table 1, suggesting vehicular emission is the main source for the enhancement of the on-road concentrations of $NO_x$, CO, $SO_2$, and BC. However, the enhancement of CO and $SO_2$ concentrations on highway were mostly less than 30%, this provides further evidence that CO and $SO_2$ concentrations on highway were dominated by regional background; and vehicular emission was not the main source.

Table 2. Estimated weighted vehicular emission factors of CO, $NO_x$, $SO_2$ and BC during mobile measurements in Hebei Province, China.

| Vehicle type | Composition | Emission factors | | | |
| --- | --- | --- | --- | --- | --- |
| | | CO[1] g km$^{-1}$ | $NO_x$[1] g km$^{-1}$ | $SO_2$[2] g km$^{-1}$ | BC[3] g km$^{-1}$ |
| Diesel bus | 20% | 1.2 | 11 | - | - |
| Medium-duty diesel vehicles | 20% | 1.5 | 6.4 | - | - |
| Light-duty gasoline vehicles | 50% | 4.4 | 1.3 | - | - |
| Heavy-duty vehicles | 10% | 1.6 | 6.6 | 0.4 | 0.11 |
| Weighted emission factor | | **2.9** | **4.8** | **0.04** | **0.01** |

[1] Shen et al., 2015; [2] Cai and Xie, 2007; [3] Lei et al., 2011

**In summary, both the comparison of on- and off-road measured concentrations and the vehicular emission factors provided the evidences**:

(1) Vehicular emission is the main source for the enhancement of the on-road concentrations of $NO_x$, CO, $SO_2$, and BC; (2) The high enhancement of $NO_x$ concentration on highway suggesting $NO_x$ on highway was mainly from vehicular emission; (3) CO and $SO_2$ concentration have up to 20% and 31% average enhancement on highway, suggesting that CO and $SO_2$ on highway were mainly from regional background.

(4) The lifetimes of CO and $SO_2$ in the atmosphere are longer than that of $NO_x$, this is likely the main reason that the concentrations of CO and $SO_2$ measured on high way were dominated by regional background, while $NO_x$ was not.

**Changes in Manuscript:** we have added detail discussions of the influence of traffic emission; please refer to the revised manuscript, from Page 15 Line 291 to Page 19 Line 391.

**Comment No. 2**. *Another concern is that the reported data may have major flaws. The extremely high $NO_x$ values are not consistent with moderate values of the other three pollutants. If the $NO_x$ data are correct, they (mean value=452 ppbv) clearly show huge impact of on-road vehicles on the measurements. However, the mean CO value is only ~1 ppm, which seems too low. Is this due to the dominance of diesel vehicles on the highways? If so, the measured black carbon would be significantly affected too by the diesel vehicles.*

**Response to Referee Comment No. 2:**

The extremely high $NO_x$ values have been reported in previous studies (Kolb et al., 2004; Westerdahl et al., 2005). As described in the response to the comment 1, we provided the evidences that the extremely high $NO_x$ values were contributed by vehicular emission, while the moderate values of CO and $SO_2$ were mainly from regional background, plus about 20-30% from vehicular emission.

In previously reported studies of on-road mobile measurements, the daily concentration of $NO_x$ could be higher than 400 ppb (Westerdahl et al., 2005). Considering that the concentration of CO on road was about 25% higher than that of off-road, and the background concentration of CO in the North China Plain was 0.4-0.7 ppm, the result of CO measured in this experiment was reasonable. Besides, the concentrations of $NO_x$ and BC measured had similar trends (Fig. 2) and correlated each other relatively well (Fig. 3). Based on this regression, we can estimated the ratio of emission factors of BC and $NO_x$ to be 0.004:1, which is close to the previously reported diesel vehicular emission factors in Beijing (0.008:1) (Wang et al., 2012). It suggests that diesel vehicular emissions had a large impact on the BC and $NO_x$ concentrations measured on highways in NCP.

[Figure]

Figure 3. The regression of the concentrations of $NO_x$ and BC measured on-road on in June 13, 2013.

**Changes in Manuscript:** we have added the discussion of the enhancements of CO and BC measured on road, please refer to the revised manuscript, from Page 15 Line 291 to Page 16 Line 324.

**Comment No. 3**. *How were the instruments calibrated? Did you make measurements off road to compare with the on-road data to check the impact of road vehicle emissions? In summary, the authors are advised to clarify these two important issues (the intended use of the on-road measurements and the data quality). In addition, the analysis and discussion of the data set should be more in-depth.*

**Response to Referee Comment No. 3:** We did make measurements off road to compare with the on-road data to check the impact of road vehicle emissions, please refer to our response to comment 1.

In previous study, we also did inter-comparison of our mobile measurement platform with a monitoring station in the campus of Peking University (Wang et al., 2009).

For this study, each time before an experiment, we did a calibration to obtain calibration curves, e. g. on June 16, 2013 (Fig. 4), and after the experiment we did another calibration and recorded the span drifts. The span drifts were less than 10%.

For example, according to the calibration on June 23, 2013, the span drifts of NO, SO₂ and CO were 29 (365, span) ppb, 9 (160) ppb, and 0.1 (7.4) ppm, respectively.

[Figure]

Figure 4. Calibration curves of NO, O₃, SO₂, and CO analyzers on June 16, 2013.

More in-depth analysis and discussion of the data set have been added in the revised version; please refer to Page 15 Line 291 to Page 19 Line 391.

**Changes in Manuscript:** we have added more detail description above the calibration method; please refer to the revised manuscript, from Page 7 Line 124 to Line 130. We also added in-depth discussion; please refer to Page 15 Line 291 to Page 19 Line 391 in the revised version.

**Referee #2:**

**Comment No. 1**. *Most seriously, I have concerns about the representativeness of the dataset, given that the measurements were only on expressways. Just because vehicle emissions aren't major sources in the regional inventory, they are very important*

*when sampling in the aggregate exhausts of other vehicles. Therefore, arguments stating that they are small regionally do not hold for these measurements. Indeed, the $NO_x$ values are argued as being influenced by vehicles – why wouldn't $SO_2$, CO, and black carbon be impacted? CO has a two-month lifetime on average, yet measurements of CO on a highway are typically much higher than, say, at a park 1 km away. It is the enhancements above the regional background that matter, and trying to separate local (road) vs. regional background measurements is not clear without extensive, other data. Did the authors drive off-expressway to examine how their measurements changed when off-expressway and (ideally) away from other vehicles (parks)? How much traffic was around the mobile laboratory? Were higher concentrations reported while in traffic jams (surrounded by cars)? The extent to which the mobile vehicle measures "expressway" versus more "regional background" levels will vary depending on the traffic density, winds, and stability. Consider the two extreme cases: 1) on a road with no other vehicles, the mobile lab will measure the regional background; 2) while stuck in traffic in a multi-lane city expressway, the measurements are clearly just vehicle emissions. How this measurement footprint changes versus location/day/traffic/wind is difficult to quantify without more measurements (see comment 3 below for more details). WFR-CHEM or FLEXPART can't model at roadway scales, of course, so there has to be some criteria applied to assess such influences. On p. 7, it is stated that traffic jams were avoided – this is rather ambiguous. How was this defined? In all this driving, there was never a traffic jam?*

**Response to Reviewer Comment No. 1:** We agree that we need to provide more evidences to justify the representativeness of our dataset for the regional background concentrations of air pollutants. Following the comments and suggestions of the reviewer, we have rewritten the manuscript and provided more evidences.

[revised manuscript text omitted]

In conclusion, in this experiment, vehicular emissions contributed less than 30% of CO and $SO_2$ measured on-road, we can use the CO and $SO_2$ measurement to study the spatial distributions of CO and $SO_2$ in NCP. Moreover, the hot spots measured near large cities were mainly from local emissions.

**Changes in Manuscript:** we have added more detailed discussions of the influence of traffic emission; please refer to the revised manuscript, from Page 15 Line 291 to Page 19 Line 391.

**Comment No. 2**. *A large number of values are reported, and too often this reads as a "data dump" with little insight besides vague generalizations. Tables comparing select case studies may be helpful, particularly highlighting the different spatiotemporal domains of the comparisons.*

**Response to Reviewer Comment No. 2:** Accept.

**Changes in Manuscript:** we have added a table (Table 3) in the manuscript to summarize the results and highlighted the different spatiotemporal domains of the comparisons in the revised paper. Please refer to the revised manuscript:

Page 21, Line 416, …on July 2 (route 1) and July 7 (route 5), respectively.

Page 21, Line 422, were regions with low emissions (Fig. 3)…

Page 21, Line 431, On June 24 (route 1), June 14 (route 4) and June 15 (route 5),…

Page 22, Line 443, …, as the same route on June 14 (Fig. 7),…

Page 24, Line 479, …different wind fields. Route 2 experienced southwest winds on June 12 and July 3,…

Page 24, Line 483, …the wind direction on route 2 in the central NCP,…

Page 24, Line 498, …the wind fields were similar on route 3 on June 13 and July 4 (Fig. 9),…

Table 3. Trip-average concentrations of $SO_2$, CO and BC and wind directions on those days discussed in the manuscript

| Route | Date | Concentrations (mean ± SD, ppb) | | | Wind directions |
|-------|------|------|------|------|-------|
| | | $SO_2$ | CO | BC | |
| Route 1 | July 2 | 4.5±2.3 | 550±240 | 5.0±2.6 | northwest |
| Route 5 | July 7 | 7.0±3.0 | 1090±320 | 6.5±2.7 | east |
| Route 1 | June 24 | 15±5.8 | 1300±330 | 8.0±1.4 | south |
| Route 4 | June 14 | 26±7.9 | 1200±230 | 6.5±1.5 | south |
| Route 5 | June 15 | 28±7.1 | 1600±370 | 7.0±1.9 | south |
| Route 4 | July 6 | 14±7.6 | 1100±450 | 4.6±1.8 | south |
| Route 2 | June 12 | 26±11 | 950±440 | 4.8±2.2 | southwest |
| Route 2 | June 18 | 27±10 | 1030±530 | 6.8±2.3 | northwest |
| Route 2 | June 25 | 27±16 | 1020±680 | 6.5±3.3 | stable |
| Route 2 | July 3 | 23±15 | 989±450 | 5.7±2.6 | southwest |

**Comment No. 3**. *Some comparisons to ground-based sites were noted (also to literature), but even these were rather trivial and limited. Comparing a transect along a nearby road to a nearby stationary site (e.g. "YC" in text), it is unclear if the mobile laboratory stopped there for a certain length of time or if it just passed by the site as part of the drive. How close to the site was the vehicle located and for what time domain for the comparisons listed (e.g. Fig. 2, p15, lines 300-310, etc.)? By knowing the mean wind, one should be able to link space-time scales at some level. Instead, just a date is given – did the authors integrate for the entire day for both measurements? Given the contamination variable contribution of vehicle exhaust in their data set, quantifying agreement is with uncertainties.*

**Response to Reviewer Comment No. 3:** Accept. We have added more information about the locations of these stationary sites in the manuscript:

1. The straight-line distance from these stationary sites to the nearest measuring roads: GC station to route 1, 3 km; QZ station to route 2, 54 km; YC station to route 2, 5 km.

2. The time domain for the comparisons: These stationary data were hour-average, and the concentrations plotted in Fig. 2 in the manuscript were hour-average results when our mobile laboratory passed by these stations.

Furthermore, the footprint maps (Fig. 5 in the manuscript and Fig. S4 in the supplement) show the wind directions and wind speeds at these sites.

**Changes in Manuscript:** we have added more information about the locations of these stationary sites, please refer to the revised manuscript, from Page 10 Line 204 to Page 11 Line 210.

**Comment No. 4**. *As the manuscript noted, concentrations are impacted by wind direction (source regions upstream), local sources (which are never really specified – just stating "urban" area), location (e.g. in the central region wind direction plays less of an influence than at the margins) and boundary layer height (e.g. p. 21, line 421-425). It is hard to interpret the variations observed in Fig. 2 – which are caused by which of the above factors and which ones dominate (and when)? Reasonable examples are provided in the following text of all of the above, but in the end, I have difficulty interpreting the importance of each of these processes in the full dataset.*

**Response to Reviewer Comment No. 4:** Agree. We discussed these four impactors in different sections in the manuscript, and to make it clearer, we added following sentences and paragraphs in the manuscript:

"Precipitation caused the low concentrations of $NO_x$ and $SO_2$ in the last experiment in NCP. And all measuring areas were affected by heavy rain, so the concentrations of $NO_x$ and $SO_2$ on all routes were low."

"We found all air pollutants measured near large cities were at high level. According to the analysis in section 3.2, they were mainly from local emissions of cities. Also, according to the estimated emission inventories, these cities were major sources of air pollution which caused the concentrations of air pollution high around them."

Also, we added a conclusion paragraph at the end of section 3.3:

"We discussed five impactors: local emission, precipitation, location, wind direction and boundary layer height. The influence of local emission reflected in the spatial distribution of concentrations (Fig. 3 in the manuscript). Hot spots were found near cities. However, for route-average results (Fig. 2 in the manuscript), local emission plays a minor role in the distribution of concentrations, because the routes were mainly in suburb. The large reduction in$SO_2$ and $NO_x$ concentration measured on July along all routes were caused by precipitation. The persistent high concentrations of pollutants in route 2 were associated with the location of it, which was surrounded by high emission areas. Besides the route 2, the different concentrations in one route in different measurements were mainly from different wind directions. And we also found one case that the concentrations changed lot between June 13 and July 4 in route 3 under the similar wind directions. Based on the model results, boundary layer height might explain the change."

**Changes in Manuscript:** we have added discussion of the five impactors, please refer to the revised manuscript, from Page 14 Line 278 to Line 280, from Page 20 Line 395 to Line 399, and from Page 26 Line 537 to Line 549.

**Comment No. 5.** *Experimental methods: what are the precisions and uncertainties in the measurements (e.g. $NO_x$ precision; CO precision and accuracy; $SO_2$ precision and accuracy; BC same thing), only $NO_x$ had an uncertainty noted (but no precision). Did the laboratory studies/calibrations done on non-driving days agree or disagree with stated manufacture specifications?*

**Response to Reviewer Comment No. 5:** Accepted. We added the precisions and uncertainties of these analyzers in section 2.1 in the manuscript (Table 4). The $NO_x$ precision was 1% and the uncertainty was less than 10%. The CO precision was 1%, and the uncertainty was less than 100 ppb. The $SO_2$ precision was 0.5%, and the uncertainty was less than 10%. The BC precision was 0.1 $\mu g\ m^{-3}$ and the uncertainty was less than 1%. These precisions were from manufacture specifications and uncertainties were from calibrations. Each time before an experiment, we did a calibration to obtain calibration curves, e.g. on June 16, 2013 (Fig. 4), and after the experiment we did another calibration and recorded the span drifts. The span drifts were less than 10%. For example, according to the calibration on June 23, the span drifts of NO, $SO_2$ and CO were 29 (365, span) ppb, 9 (160) ppb and 0.1 (7.4) ppm. We added this information in the manuscript.

Table 4. Precisions and uncertainties of air pollutants analyzers used in our experiments

|  | $NO_x$ | CO | $SO_2$ | BC |
|---|---|---|---|---|
| Precision | 1% | 1% | 0.5% | 0.1 $\mu g\ m^{-3}$ |
| Uncertainty | <10% | <100 ppb | <10% | <1% |

| Time resolution | 30 s | 40 s | 120 s | 6 s |
|---|---|---|---|---|

**Changes in Manuscript:** we have added more detailed information of the precisions and uncertainties of these analyzers, please refer to the revised manuscript, from Page 6 Line 115 to Line 130.

**Comment No. 6.** *To what extent was the data coverage for the 5 campaigns? 100%? Rain is mentioned on p. 8, line 152 – what about other data dropouts (if they existed)?*

**Response to Reviewer Comment No. 6:** Accept. The major reasons for data missing were the computer crashing and rain. Rain caused the missing data on route 3, route 4 and route 5 in experiment 3, and on route 5 in experiment 5. Computer crashing caused the missing data on route 4 and route 5 in experiment 2 (Table 5). Also, the computer crashing caused data missing during every trip (Table 6). We added the information in the manuscript.

Table 5. Measurement data and routes in the 5 campaigns

| Date | Route 1 | Route 2 | Route 3 | Route 4 | Route 5 |
|---|---|---|---|---|---|
| June 11–June 15 | June 11 | June 12 | June 13 | June 14 | June 15 |
| June 17–June 20 | June 17 | June 18 | June 20 | - | - |
| June 24–June 25 | June 24 | June 25 | - | - | - |
| July 2–July 7 | July 2 | July 3 | July 4 | July 6 | July 7 |
| July 11–July 15 | July 11 | July 12 | July 14 | July 15 | - |

Table 6. Data coverage of every trip

| Date | Start time | End time | Data coverage | Reason for data missing |
|---|---|---|---|---|
| June 11 | 10:00 | 14:00 | 67% | Computer crashing |
| June 12 | 10:23 | 14:23 | 92% | Computer crashing |
| June 13 | 10:15 | 14:00 | 100% | |
| June 14 | 9:55 | 13:55 | 100% | |
| June 15 | 9:58 | 14:14 | 99% | Computer crashing |
| June 17 | 10:40 | 14:20 | 78% | Computer crashing |
| June 18 | 9:55 | 13:35 | 80% | Computer crashing |
| June 20 | 10:15 | 13:05 | 85% | Computer crashing |
| June 24 | 10:50 | 14:10 | 66% | Computer crashing |
| June 25 | 10:20 | 14:20 | 83% | Computer crashing |

| | | | | |
|---|---|---|---|---|
| July 2 | 10:23 | 14:15 | 72% | Computer crashing |
| July 3 | 10:27 | 13:35 | 72% | Computer crashing |
| July 4 | 10:10 | 11:46 | 100% | |
| July 6 | 9:58 | 14:22 | 98% | Computer crashing |
| July 7 | 10:12 | 14:46 | 74% | Computer crashing |
| July 11 | 10:22 | 14:30 | 66% | Computer crashing |
| July 12 | 10:10 | 14:07 | 84% | Computer crashing |
| July 14 | 10:13 | 11:42 | 100% | |
| July 15 | 10:12 | 15:05 | 88% | Computer crashing |

**Changes in Manuscript:** we have added information of data missing, please refer to the revised manuscript, from Page 8 Line 160 to Line 164, and the supplement Table S1 and Table S2.

**Comment No. 7.** *p. 12, "low levels of $SO_2$": 10 ppbv $SO_2$ is not low, perhaps compared to the past in China. But I'm not sure how this mobile lab study over a very limited time/space can verify that $SO_2$ has decreased and confirmed policy desulphurization devices. No comparison to past $SO_2$ measurements in these locations is noted, so this seems unsubstantiated from this study.*

**Response to Reviewer Comment No. 7:** Accept.

**Changes in Manuscript:** we deleted the "low levels of $SO_2$" and related discussions in the manuscript. Please refer to the revised manuscript, from Page 13 Line 254 to Line 255.

**Comment No. 8.** *Some data should be made available to the community on a website, not "on request", given the political sensitivity of Chinese air pollution data. What metadata will be provided, e.g. GPS, vehicle information (speed), met data, etc.?*

**Response to Reviewer Comment No. 8:** Agree. We are happy to provide our data on a website. We had a website for this purpose during CAREBEIJING Campaign. However, this site is closed due to lake of fund. We are working on it and hoping to reinstall the web site in this year. We will provide data of GPS, vehicle speed, meteorology, concentrations of air pollutants.

**Changes in Manuscript:** we have added more information about data availability; please refer to the revised manuscript, from Page 28 Line 570 to Line 571.

**Comment No. 9.** *I like the FLEXPART trajectory attribution, but again this is a qualitative comparison as the upwind values were not measured. So while it makes sense that northerly winds off the mountains will decrease values nearby, I don't think this was particularly profound.*

**Response to Reviewer Comment No. 9:** Agree. This argument was based on a general situation that the air quality in the north border of the North China Plain was clean, because the emissions of air pollutants were low in that area (Fig. 3 in the manuscript). To support this argument, we collected the data of air pollutants concentrations measured on the Huairou site (116.63 $^O$E, 40.32 $^O$N) on July 2, 2013 from State Control Monitoring Sites database (http://www.mep.gov.cn/hjzl/). The average concentrations of CO, $NO_2$, and $SO_2$ from 9:00 to 15:00 on that day were 0.3 ppm, 3.4 ppb, and 3.3 ppb, respectively. It indicates that the concentrations of air pollutants in the north border of NCP were low.

**Changes in Manuscript:** we have added explanation of the clear air from north of NCP, please refer to the revised manuscript, from Page 23 Line 458 to Line 460.

**Reference**

[revised manuscript text omitted]